# The effects of aging and an episodic specificity induction on spontaneous task-unrelated thought

**Magda Jordão**[1¤]*, **Maria Salomé Pinho**[1], **Peggy L. St. Jacques**[2]

**1** Center for Research in Neuropsychology and Cognitive and Behavioural Intervention, Faculty of Psychology and Educational Sciences, Univ Coimbra, Coimbra, Portugal, **2** Department of Psychology, University of Alberta, Edmonton, Canada

¤ Current address: Faculdade de Psicologia e Ciências da Educação da Universidade de Coimbra, Coimbra, Portugal

* magda.jordao@gmail.com

**Data Availability Statement:** The dataset generated in the current study and the vigilance tasks created are available at (https://osf.io/cn28e/).

## Abstract

When voluntarily describing their past or future, older adults typically show a reduction in episodic specificity (e.g., including fewer details reflecting a specific event, time and/or place). However, aging has less impact on other types of tasks that place minimal demands on strategic retrieval such as spontaneous thoughts. In the current study, we investigated age-related differences in the episodic specificity of spontaneous thoughts using experimenter-based coding of thought descriptions. Additionally, we tested whether an episodic specificity induction, which increases episodic detail during deliberate retrieval of events in young and older adults, has the same effect under spontaneous retrieval. Twenty-four younger and 24 healthy older adults performed two counterbalanced sessions including a video, the episodic specificity or control induction, and a vigilance task. In the episodic specificity induction, participants recalled the details of the video while in the control they solved math exercises. The impact of this manipulation on the episodic specificity of spontaneous thoughts was assessed in the subsequent vigilance task, in which participants were randomly stopped to describe their thoughts and classify them as deliberate/spontaneous. We found no differences in episodic specificity between age groups in spontaneous thoughts, supporting the prediction that automatic retrieval attenuates the episodic specificity decrease in aging. The lack of age differences was present regardless of the induction, showing no interactions. For the induction, we also found no main effect, indicating that automatic retrieval bypasses event construction and accesses pre-stored events. Overall, our evidence suggests that spontaneous retrieval is a promising strategy to support episodic specificity in aging.

## 1. Introduction

Age-related differences are typically attenuated when the amount of self-initiated processing is minimal [1]. Currently, it is unclear if this is also the case for spontaneous thoughts which are

**Funding:** This research was supported by a PhD grant (SFRH/BD/103338/2014) provided to MJ by the Fundação para a Ciência e Tecnologia (https://www.fct.pt/index.phtml.en) through the Human Capital Operating Programme supported by the European Social Fund and the Ministry for Science, Technology and Higher Education of Portugal. The funders had no role in study design, data collection and analysis, decision to publish, or preparation of the manuscript.

**Competing interests:** The authors have declared that no competing interests exist.

not self-initiated and come into awareness with reduced intentionality and effort [2,3]. Spontaneous thoughts are frequent daily occurrences [4] and play an important role in planning, creativity, and memory consolidation [5]. Past-oriented spontaneous thoughts, in particular, are at least as frequent as deliberate memories [6] and contribute to maintaining a sense of time and personal continuity [7]. Aging research has been strongly focused on the occurrence of spontaneous thoughts in association with mind wandering research, a phenomenon closely related to spontaneous thought. A recent meta-analysis found a reduced frequency of spontaneous thoughts in older compared to younger adults but identified methodological moderators suggesting that this effect may be partially due to the experimental paradigms most commonly used [8]. This is in line with a recent review of the effects of healthy and pathological aging on spontaneous thoughts that suggests that the decrease in spontaneous thoughts in healthy aging is absent when the experimental conditions include meaningful cues that activate the spontaneous retrieval route [9]. The impact of aging on qualitative aspects of spontaneous thoughts, in particular episodic specificity, has been much less explored. Episodic specificity refers to the amount of detail about an event (e.g., what, when and where it happened), and is typically reduced in aging during deliberate retrieval of autobiographical memories and imagined events (for a review, see [10]). However, recent research has shown that an episodic specificity induction (ESI) that involves training in recollecting details of past events can increase episodic specificity in aging [11] by targeting episodic retrieval processes that support the construction of event representations [12]. In the current study we analyzed the impact of aging on episodic specificity and used the ESI to determine whether constructive processes contribute to the episodic specificity of spontaneous thoughts.

## 1.1. Aging and spontaneous thought

A growing number of studies have examined the influence of aging on spontaneous thought frequency as the result of increasing interest in mind wandering. Mind wandering (MW) describes a shift of attention from an external task to internal contents [13]. Although MW can sometimes be deliberate [14], it is typically characterized as spontaneous thought [2], such that spontaneous MW is three times more frequent than deliberate MW across age groups [15]. Older adults typically have a reduction in MW frequency [16], but in a recent meta-analysis we found that methodological differences related to reporting mode, response options, task difficulty, and socio-demographic variables contributed to the pattern of age-related differences [8]. Importantly, when controlling for these methodological variables, we found no evidence of age-related differences in the frequency of spontaneous task-unrelated thoughts [17]. Similar results have been found in naturalistic studies using diary and experience sampling methods [18,19].

Research on involuntary autobiographical memory (IAM) also contributes to the understanding of age effects in spontaneous thoughts. Involuntary memories come to mind effortlessly and without a previous retrieval attempt [20], and are thus a type of spontaneous thought. Several studies have demonstrated a lack of age-related differences in IAM frequency as measured by inventory [21], questionnaire [22,23], or diary studies [24]. Would the lack of age-related changes in the frequency of MW and IAM extend to qualitative aspects of spontaneous thought? In the next section, we explore this question for episodic specificity.

## 1.2. Aging and episodic specificity

Episodic specificity refers to the degree to which a content includes specific event information ("what", "when" and "where") and experiential detail [25]. Aging leads to a decrease in episodic specificity during deliberate recall of personally experienced past events (i.e.,

autobiographical memories; e.g., [10,24,26–28]) and future events (e.g., [29–30]). The age-related decrease in episodic specificity has been shown using different measures, ranging from the simplest distinction between a specific versus general event to more complex classification systems such as the objective assessment of participants' descriptions using the Autobiographical Interview (AI; [26]) and the *Test Episodique de la Mémoire du Passé lointain autobiographique* (TEMPau; [31]). Both of these objective assessments of episodic specificity are less susceptible to age-related biases than detail ratings provided by participants (in which different age patterns are found; [29,32]). In terms of mechanisms, the age-related decrease in episodic specificity is associated with reductions in strategic elaboration [33] and effortful processes such as executive functions, both in past [27,34] and future thoughts [30,35]. Thus, we would expect age-related differences in episodic specificity to be reduced when retrieval is less reliant on strategic processes, such as in spontaneous thoughts. In fact, a key difference between deliberate and spontaneous retrieval is that the former is more effortful as shown, for example, by slower retrieval times [36] and by the involvement of brain regions associated with monitoring and cognitive control [37]. Effortful retrieval processes are associated with generative retrieval, that is, a strategic process of search that begins at the most general level of knowledge about oneself and by successive iterations accesses a specific event [38]. Alternatively, event representations about the past [39] and future [40] may be accessed effortlessly in a direct or associative fashion [38,41], purportedly based on a process of cue-item discriminability by which a distinctive cue isolates a specific event by automatic spreading activation [20,42]. Spontaneous retrieval by definition involves more direct than strategic search processes, and by its effortless nature, should make it easier for older adults to access event specific information.

In contrast with the wealth of data for deliberate retrieval, less is known about age-related changes in episodic specificity for spontaneous retrieval. The current evidence suggests that aging does not reduce episodic specificity for spontaneous thoughts. Schlagman and collaborators investigated IAMs in diary studies in which younger and older adults recorded every memory that came to mind and classified them as referring to a single, extended, or repeated events, and found no age-related differences [24,43]. However, these findings were based solely on self-report, which could introduce biases (e.g., participants classify IAMs as more specific when asked to report only memories versus any type of content; [44]). In sum, it is necessary to examine age-related differences in episodic specificity based on the independent assessment of participants' descriptions, which will also facilitate comparisons between age-related differences in spontaneous and deliberate retrieval [26,45].

## 1.3. Episodic specificity induction

Recent research has shown that episodic specificity can be increased experimentally, in both younger and older adults, using an episodic specificity induction (ESI; [12]). The ESI increases episodic detail by leading to a specific retrieval orientation that facilitates the construction of specific episodes, that is "the assembly of a mental scenario bound in space and time with details related to settings, people, and actions" [46, p. 2]. It is important to note that the term "construction" is frequently used in memory literature to refer to different concepts and/or processes [47]. Here, we are consistent with other ESI studies and define construction by focusing on the process of binding the different types of episodic details that constitute an event.

The ESI consists of a brief training based on the cognitive interview [48] that focuses on the recall of specific details. For example, Madore, Gaesser, & Schacter [11] used the ESI to look at event construction in memory and imagination. In this study, participants watched a brief video and subsequently recalled it. During the ESI, the experimenter asked participants to

recall the video focusing on the details (objects, people, and actions) using pre-determined questions. During the control condition, participants were instructed to focus on their general impressions about the video. Following the ESI or control condition, participants were asked to describe memories and future thoughts. More episodic details (as measured by the Autobiographical Interview coding; [26]) were recalled in both memories and future thoughts following the ESI compared to the control condition. Several studies have replicated these finding in other deliberate tasks (creative thinking in [49], problem-solving in [50]) and using a measure of scene construction instead of episodic detail [51]. Thus, the ESI is a robust method to target event construction and increase episodic specificity.

Whether the ESI effect will also impact episodic specificity in spontaneous thoughts is currently unknown. By nature, spontaneous thoughts do not involve the type of goal-directed and deliberate nature of tasks that are typically influenced by ESI. Despite the lack of intention and seemingly ease with which spontaneous thoughts come to mind, spontaneous thoughts might still rely on event construction because they share the same episodic memory system and differ only in the effort required at retrieval (for a review, see [52]). Specifically, during spontaneous thoughts cues "activate event-relevant units, or nodes, in the network and deactivate irrelevant units that would otherwise interfere with the construction of the memory" ([20], p. 106), thus, reducing the requirement for effortful search processes. Alternatively, spontaneous thoughts might rely more on direct retrieval processes that imply the existence of pre-stored event representations, independent of event construction [39]. Supporting this idea, spontaneous future thoughts are characterized by relatively short reaction times ([40,53], for a review). According to this perspective, the ESI should not affect spontaneous thoughts. Thus, investigating the influence of the ESI will reveal whether spontaneous thoughts involve event construction. In practice, it will indicate whether the ESI is useful to increase episodic specificity in spontaneous thought.

### 1.4. The present study

In the present study we examined the effects of aging on episodic specificity of spontaneous thoughts reported by healthy younger and older adults in a laboratory task. In two sessions separated by approximately a week, we used either the ESI or a control induction followed by a vigilance task to elicit spontaneous thoughts, which were audio-recorded at random stops and later analyzed by independent coders for episodic specificity. We had two main aims. First, we investigated whether the lack of age-related differences in the episodic specificity of IAMs would generalize: (i) in a lab-task, (ii) from past to spontaneous thoughts in general, and future thoughts particularly, and (iii) when episodic specificity was assessed by independent coders. The lack of age-related differences in our study would indicate that age effects in episodic specificity are diminished in spontaneous retrieval. Second, we examined whether the episodic specificity of spontaneous thoughts depends upon the deliberate involvement of event construction by comparing the influence of the ESI to a control induction procedure prior to the elicitation of spontaneous thoughts [12]. Additionally, given the novelty of the present approach and the scarce evidence on the topic, we analyzed several phenomenological variables including emotional valence and arousal (that have been shown to interact with specificity in deliberate retrieval; [28,54]), visual/verbal imagery, and detail based on subjective ratings.

## 2. Materials and methods

This study was approved by the Ethics Committee of the Faculty of Psychology and Education Sciences of the University of Coimbra. Oral consent for participation was obtained prior to

data collection as approved by the Ethics Committee. The oral format was chosen due to the reluctance of some older participants to provide their written signature, and was obtained for every participant by the first author. The younger participants were undergraduate Psychology students that chose to participate for extra credit. Older adults did not receive compensation for participating.

## 2.1. Sample

To determine the sample size necessary to identify an ESI effect, we reviewed previous studies with younger and older adults. For memories and imagined scenarios the effect ranges from .62 to .78 [11,50,55]. Based on an a priori power analysis, considering an effect of $d$ = .60, power = .80 and a two-tailed repeated measures test we determined a sample of 24 participants for each age group [56]. This sample size is adequate to identify large overall age-differences in specificity ($d$ > .80, power = .80, one-tailed test) similar to studies investigating memory and imagination (e.g., [11,50]). This should ensure that when overall age-related differences in episodic specificity in spontaneous thoughts are similar to those found in deliberate thoughts, we will be able to identify them.

Participants were excluded if they reported a history of neurological or psychiatric diagnosis, and/or moderate to severe depressive symptomatology, based on a total score of more than 18 assessed using the Beck Depression Inventory II (BDI-II [57,58]). Older adults also performed a cognitive function test and no participant showed evidence of cognitive decline (based on Portuguese norms, cut-off 2 standard deviations below the mean for age and education level; Montreal Cognitive Assessment, MoCA; [59,60]). Thirteen participants (10 younger and 3 older adults) were excluded based on a history of neurological or psychiatric diagnosis, and 5 participants (4 younger and 1 older adult) based on moderate to severe depressive symptomatology. The final sample comprised 24 younger adults [22 women, mean age in years ($M$) = 20.21, $SD$ = 2.75, range 18–30] and 24 older adults (22 women, $M$ = 67.58, $SD$ = 3.92, range 61–77). There was no evidence of age-related differences in the number of years of education (based on $U$ Mann–Whitney, $Z$ = - 0.95, $p$ = .34) between younger ($M$ = 13.98, $SD$ = 1.62) and older participants ($M$ = 13.17, $SD$ = 4.54).

## 2.2. Design

The study used a quasi-experimental design, with type of induction (episodic specificity induction, control induction) as a within-subjects variable and age group (younger, older) as a between-subjects variable. Our main dependent variables were the frequency of spontaneous task-unrelated thoughts and their episodic specificity, as measured by an overall specificity score, a specific events measure and strictly episodic score (see section 2.6.2 for more details on the episodic specificity measures).

## 2.3. Procedure

The experimental procedure is represented in Fig 1. Participants attended two sessions, approximately 7 days apart ($M$ = 7.13, $SD$ = 1.30). Both sessions included an initial induction procedure (episodic or control), in which a video was presented, followed by a vigilance task to elicit spontaneous thoughts. Task presentation was counterbalanced in all eight possible combinations for order of induction (control, episodic specificity), video (version A, version B) and vigilance task (version A, version B), to rule out order effects. There was an exception to the full counterbalance for one participant in the older adults' group. In this case, the order of presentation of the videos was switched, so that there is one more participant in this group with one of the two possible orders for video presentation. Importantly, this did not affect the

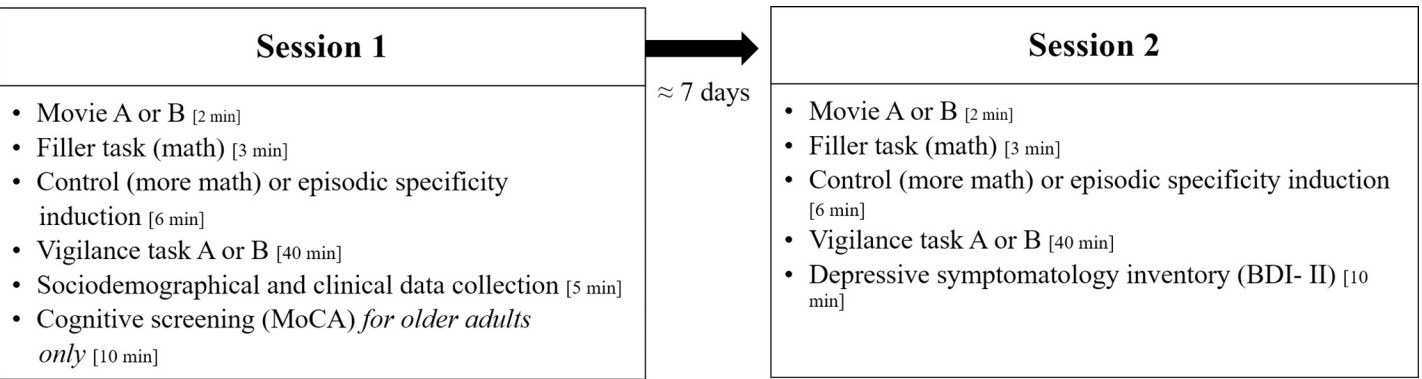

**Fig 1. Experimental procedure.** MoCA stands for Montreal Cognitive Assessment ([59]; European Portuguese version, [60]) and BDI-II stands for Beck Depressive Inventory II ([57]; European Portuguese version, [58]).

counterbalance of other variables (i.e., vigilance task version and control/episodic specificity induction). Thus, it was unlikely that this case would impact the results and it was included in the final sample.

At the end of the first session, we collected sociodemographic and clinical information, and older adults were administered the cognitive screening test. At the end of the second session, both groups filled in a depressive symptomatology inventory and were asked about what they thought the aim of the experiment was. No participant mentioned that the experiment aimed to analyze specificity.

## 2.4. Episodic specificity and control induction

For the episodic specificity and control induction we followed the procedure applied previously [11,49,50]. Participants were assigned to two sessions, beginning with a 2-minute video, which we asked participants to watch attentively. The video was different in each session, but both depicted a man and a woman performing actions in a kitchen (e.g., taking a detergent box outside of a paper bag, putting a bunch of flowers into water). This was followed by a 3-minute filler task (addition/subtraction math problems). Then, in the ESI, participants were asked to create a mental image and describe the details of the surroundings, people and actions depicted in the video as completely as possible. In contrast, in the control induction, participants were asked to solve more math problems [49], as this has been identified as the most neutral control condition (see [11]). The instructions were translated from English to European Portuguese by one of the authors (MJ) and edited for clarity with two Portuguese native speakers with research experience.

## 2.5. Vigilance task

After the induction procedure, participants performed a vigilance task designed to elicit spontaneous task-unrelated thoughts (sTUTs). Prior to starting the task participants assessed their motivation level on a scale from 1 (not motivated at all) to 5 (completely motivated) to perform the task. By asking for this assessment before the task, we aimed to avoid the influence of perceived performance on these ratings [61].

The vigilance task included 159 trials consisting of 72 words or 87 five-dot sequences (".... .") that were presented randomly. Each stimulus was on screen for 3 seconds, followed by a 3 seconds inter-stimuli interval (ISI, as represented in S1 File). The total duration of the vigilance task without probes was 15.9 minutes. Stimuli were presented either in black or yellow and

participants were asked to say "yes" out loud when a yellow stimulus was presented (target), which were recorded by an experimenter. Only three of the stimuli were presented in yellow (1.9% of the stimuli, similar to reference [36]). Stimuli were displayed in 64-point bold Arial on a 1366 pixels (width) x 768 pixels (height) screen, using E-Prime 2.0 (Psychology Software Tools, Pittsburgh, PA).

During the vigilance task, randomized probes were presented in intervals of 54, 78 and 108 seconds (or 9, 13 or 18 trials), similar to previous studies (e.g., 52.5 to 105 seconds, see reference [62]). When probes were presented, participants were asked to stop and "describe out loud everything you can about what was on your mind immediately before you saw this screen" to an audio recorder. To make sure the participants reported everything that came to mind, a standardized prompt was used after every description ("Can you describe anything more about that thought? I want to know all the details that you thought about"). Immediately after, participants answered additional questions presented verbally by the experimenter, including ratings for spontaneity, triggers, temporality, visual imagery, valence, arousal and detail. For spontaneity, participants used a scale from 1 = I wasn't trying to bring this to my mind at all, to 7 = I tried very hard (as in e.g., reference [63]). For triggers, participants were asked to indicate whether the content was triggered by an external stimulus, and if so, what stimulus. For temporality, participants classified a thought as past, present or future-oriented if it was related to something occurring before, during or after the task, respectively, and atemporal if the thought lacked a temporal orientation (following reference [64]). For visual imagery, participants classified their thoughts as predominantly verbal or visual (following reference [65]). For valence, arousal and detail, participants used a 1 to 7 scale (from very unpleasant to extremely pleasant, not intense at all to extremely intense and not detailed at all to extremely detailed). After the completion of the vigilance task, participants were asked to indicate their level of concentration and the difficulty of the task (from 1 = not concentrated/ difficult at all to 5 = extremely concentrated/difficult).

There were two versions of the vigilance task for each session. Each version included 72 different words which were included in the task to facilitate spontaneous retrieval (for the role of meaningful stimuli in spontaneous thought and aging see [8]). These words were selected from an extensively studied word pool that were matched in terms of valence, arousal [66], frequency, concreteness and imageability ([67], see S1 File for the list of words and statistical comparison of each version). The cue frequency was approximately one cue-word every 13.33 seconds ($SD$ = 3.85).

This vigilance task was based on our previous adaptation [17] of a task created by Schlagman and Kvavilashvili [36]. Our earlier study found no evidence of age differences in MW frequency [17], suggesting that the adaptation controlled for confounding factors, such as possible age-related differences in the use of pre-determined and forced-choice response options to describe mental contents [8]. In the current study, we made three additional improvements. First, we increased the number of probes to 12, increasing the ability to capture sTUTs. Second, we made the content between each probe comparable, by presenting the same number of word cues with equivalent characteristics in each probe interval. Third, we used an ISI of 3 seconds following the presentation of each stimulus, to facilitate catching spontaneous thoughts. The 3 second ISI does not require a thought to be maintained for a long period of time in order to be caught by the probe.

## 2.6. Experimenter coding

**2.6.1. Type of thought.** To classify thoughts elicited during the vigilance task, we followed a family resemblances view of MW [68]. According to this approach, MW is a naturally

heterogeneous concept that includes sTUTs, and the main concern of experiments should be to specify the type of MW being assessed. Here, we determined if thoughts were related or unrelated to the task based on independent coders' assessment of the descriptions provided by participants. We started by asking participants to classify task-relatedness on a 1 to 7 scale (not at all related to completely related), but a preliminary analysis of the responses suggested that this question was confusing. Specifically, some participants would automatically classify a thought as task-related if it was triggered by a stimulus presented in the task, irrespective of the content being related to the task or not, thereby confounding the stimulus-(in)dependency and task-(un)relatedness dimensions. Based on this observation we decided to code task-relatedness using independent assessment by coders. An example of a task-related thought from our sample was "I was thinking that my reaction time to the yellow [target] is increasing as the task goes on". Examples of thoughts unrelated to the task are presented in Fig 2.

Additionally, independent coders identified two types of sTUTs that were not suitable for the episodic specificity analysis: external distractions and earworms. First, we defined external distractions (EDs) as "sensory perceptions/sensations irrelevant to the current task" ([69], p. 371), in which environmental features capture the participant's attention (as in [69,70]). This included thoughts such as "My feet are getting cold" and "The ladies [outside] are talking too loud". Second, we separately defined earworms or involuntary musical imagery in which participants described having only music in mind [71]. There is evidence that musical memories rely on a different cognitive system and show distinct age-related patterns (e.g., [72]), suggesting possible interactions with aging also in spontaneous retrieval. For this reason, we analyzed these spontaneous thoughts separately.

**2.6.2. Episodic specificity.** Unlike previous studies using the ESI with older adults [11,50,55], we were not able to assess specificity based on the number of internal and external details [26]. This was due to the nature of the descriptions provided by participants in which much of the information related with time and place was provided implicitly. Take the

| Coding | Description (45) | Examples |
|:---:|:---:|:---:|
| 0 | General knowledge about a theme | *I like thunderstorms… There are people who are afraid of it, but I'm not…* |
| 1 | Repeated or extended event **not** situated in time and place **or** atemporal scenario | *I was thinking about my daughter, who is scared of any little bug – whenever she sees one, she starts to scream "mum, there's a bug there!" right away* |
| 2 | Repeated or extended event situated in time and place | *When I saw the word tank, I recalled that in my childhood I used many times to bath in a big tank my godmother had in her property…* |
| 3 | Specific event situated in time (less than 24h) and place with **no** additional details | *I was thinking that I deserve when arriving at my city to have a hot chocolate in the Luna coffee* |
| 4 | Specific event situated in time (less than 24h) and place with additional details (such as feelings, perceptions, thoughts, or visual imagery) | *I remembered my cousin - the other day I went to visit him, I recall playing with him and then trying to put him to sleep, he was on the top of a snooker table with several toys around him* |

**Fig 2. Examples of task-unrelated thoughts coded for specificity based on the TEMPau (the descriptions of each category are adapted from [45]).**

following example: "I was thinking that my roommate let olive oil burn and then we had to be lightening up candles to see if that smell went away". Here, there is no explicit mention of a specific time and place, and, thus, this information would not be scored based on the number of internal and external details [73]. However, the description does imply a specific event, in contrast with descriptions in which time and place details are not mentioned such as: "I was thinking about my brother, I imagined his image and his way of being". Additionally, the descriptions were usually short. In contrast, participants typically provide a narrative with a beginning, middle and end when asked to explicitly recall memories, and are thus more likely to naturally mention details related to time and place. To better capture episodic aspects of spontaneous thoughts we used the coding scheme of the TEMPau [45], which focuses on the nature of the event described (repeated or extended in time, with or without a place) while still enabling the identification of situations in which additional detail (such as feelings or visual imagery) is provided. Importantly, the TEMPau can capture age-related differences in episodic specificity for voluntary thoughts [45]. One minor change was introduced to the coding scheme to account for atemporal scenarios (e.g., "I saw many refugees in a small boat, struggling"). Atemporal scenarios were coded 1 although they did not include time information (see S2 File for instructions) in order to distinguish these more detailed descriptions from general information statements (in line with reference [65]). In Fig 2, we present examples of the adapted TEMPau categories from the present data.

To examine episodic specificity we calculated overall and strictly episodic scores, based on the TEMPau [45]. The overall score includes all instances in which participants described an event, either specific or generic (levels 1 to 4 in the TEMPau), which allowed us to characterize thoughts associated with events irrespective of whether they referred to a unique experience or not. This is important following the idea that events are key to mental time travel and provide a better contrast to semantic memory than the unique occurrences emphasized by episodic memory [74]. In contrast, the strictly episodic score includes only specific events described with detail (level 4 in the TEMPau). The presence of phenomenological detail associated with a specific event is considered diagnostic of the degree of episodicity and reliving ([45], based on [75,76]), which was a central focus of the current study. We included one additional measure of episodic specificity to capture thoughts referring to specific events with and without detail (levels 3 and 4 in the TEMPau; based on [24]). This is a widely accepted definition of episodic specificity that has been shown to adequately distinguish psychopathological memory changes [77]. Additionally, including the same measure used as a previous diary study on IAMs [24] is important to assess whether the null age-related effect generalizes when tested in the laboratory environment.

Finally, we also investigated differences in episodic specificity using all of the categories distinguished in the TEMPau coding scheme [45]. These included: general knowledge, repeated/extended event not situated in time and place, repeated/extended event situated in time and place, specific event situated in time (<24h) and place without additional details, and specific event situated in time (<24h) and place with additional details. This was an exploratory analysis deemed important by the scarcity of studies analyzing the specificity of sTUTs and aging.

The complete coding process is summarized in Fig 3. As shown, we focused on identifying sTUTs and determining episodic specificity, by distinguishing task-unrelated thoughts by TEMPau level [45]. As represented, we used the TEMPau levels to determine the strictly episodic score (level 4), the episodic event score (level 3 and 4) and the overall score (level 1 to 4). This coding process provides a measure of the frequency of specific events as reported by participants, and better allows for comparisons with previous research [24,45].

**2.6.3. Interrater reliability.** Two coders (MJ and a Psychology student) categorized each thought record according to whether: (1) they were a case of external distraction, earworm or

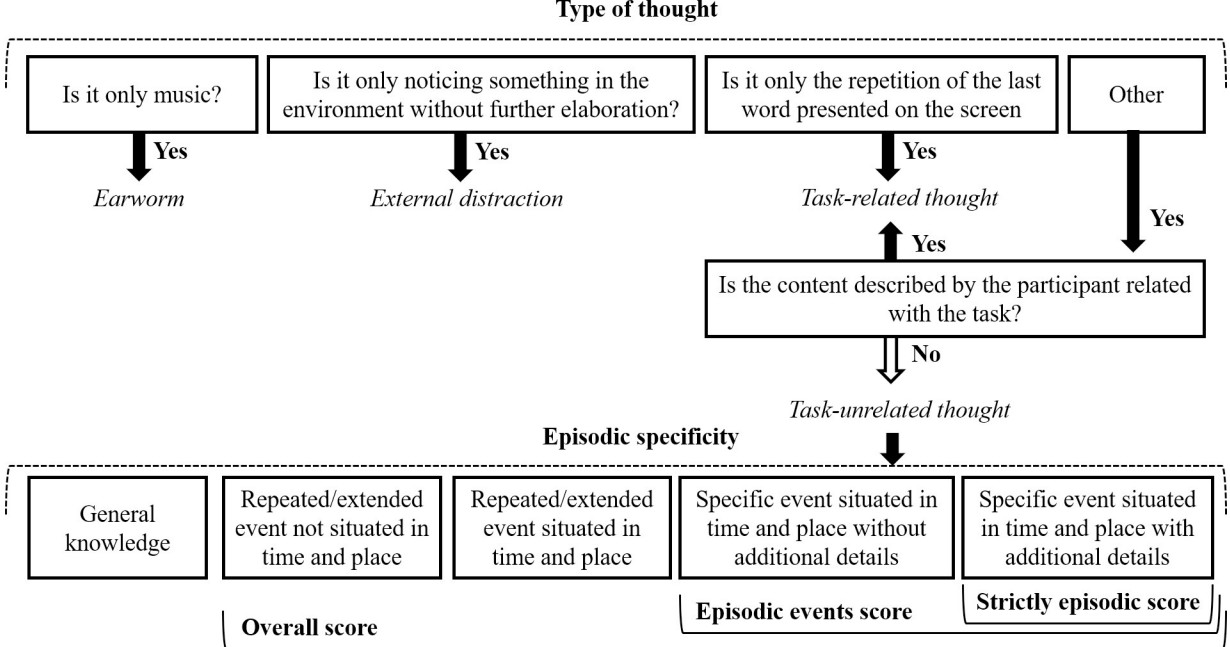

**Fig 3. Summary of the experimenter coding process for thoughts classified as spontaneous by the participants (the TEMPau categories are based on [45]).**

none of those two, (2) they were task-related or unrelated, (3) episodic specificity. Situations in which participants reported they had nothing on their mind, and/or non-spontaneous thinking were not coded, but instead were excluded from further analyses. Coders were blind to the experimental conditions, and all coders but MJ were blind to the hypotheses; however, participant age could sometimes be inferred from the audio record. Disagreements were analyzed by a third independent coder (a Psychology student) who was also blind to experimental condition and hypotheses. The interrater reliability was good (for OAs; *Kappa* = .73) or very good (for YAs; *Kappa* = .80) for identifying external distractions and earworms, very good (*Kappa* = .87 for OAs; *Kappa* = .89 for YAs) for task-(un)relatedness, and good (weighted *Kappa* = .79 for OAs; weighted *Kappa* = .72 for YAs) for episodic specificity.

## 3. Results

### 3.1. Statistical analyses

We were interested in possible null effects for sTUTs frequency and episodic specificity. Thus, for these variables we followed null results in the frequentist analyses with Bayesian analyses. These were performed with the JASP software [78] and, if not indicated otherwise, are based on the JASP default settings (fixed effects with r scale prior width of 0.5 for repeated-measures ANOVAs). For the frequentist analyses, we used an alpha level of 0.05 and performed all tests with the Statistical Package for the Social Sciences (SPSS) software, version 22.

### 3.2. Self-rated motivation, concentration and task difficulty

We investigated motivation, concentration and task difficulty ratings separately in 2 (age group: young, older) × 2 (type of induction: ESI, control) mixed ANOVAs (see variable description by group and induction in Table 1). We found no main effects or interactions (*p*'s > .11), except for concentration, in which older adults reported greater concentration levels

**Table 1. Mean ratings (standard deviation) of motivation, concentration and difficulty by age group in the episodic specificity induction (ESI) and in the control induction.**

|  |  | ESI | Control |
|---|---|---|---|
| Motivation | Younger | 4.17 (0.70) | 4.21 (0.78) |
|  | Older | 4.42 (0.93) | 4.46 (0.72) |
| Concentration | Younger | 3.75 (1.03) | 3.67 (0.82) |
|  | Older | 4.29 (0.75) | 4.29 (0.75) |
| Difficulty | Younger | 1.58 (1.10) | 1.46 (0.72) |
|  | Older | 1.13 (0.45) | 1.42 (0.78) |

($M$ = 4.29, $SD$ = 0.72) than younger adults ($M$ = 3.71, $SD$ = 0.72) across type of induction, $F(1,46) = 7.69$, $MSE = 8.17$, $p = .008$, $\eta_p^2 = .14$.

## 3.3. Type of thought

Overall, a total of 1152 probes was presented, 576 for each age group. We started by analyzing spontaneity based on the ratings given by participants. Thoughts rated from 1 to 3 were categorized as spontaneous, 4 as undecided, and 5 to 7 as deliberate. The majority of the thoughts were spontaneous (84% in YA and 86% in OA). Younger adults reported spontaneous thoughts in 298 probes (52%) and OAs in 315 probes (55%). For YAs, these probes included 14 earworms, 6 EDs, 53 task-related thoughts and 225 sTUTs. For OAs, these probes included no earworms, 25 EDs, 83 task-related thoughts and 207 sTUTs. The mean frequency and standard deviation of each type of spontaneous thought by type of induction and age group is presented in Table 2. Inspection of the frequency distributions of earworms, external distractions and task-related thoughts revealed that they were not normal (Shapiro-Wilk test, $p < .001$) and variances were mainly heterogeneous ($p \leq .001$, except task-related thoughts in ESI, $p = .310$), thus we used the Mann-Whitney $U$ test to test for differences due to age-group. Earworms were more frequent for younger than older adults in both the control ($Z = -2.34$, $p = .02$) and ESI ($Z = -2.59$, $p = .01$). Younger adults also experienced significantly fewer EDs than older adults in the control induction ($Z = -2.54$, $p = .01$).

## 3.4. Spontaneous TUTs frequency

To examine the frequency of sTUTs we performed a 2 (age group: young, older) × 2 (type of induction: ESI, control) mixed ANOVA. As expected, there was no main effect for age group ($p = .695$), type of induction ($p = .765$), or their interaction ($p = .857$).

**Table 2. Mean number (standard deviation) of spontaneous earworms, external distractions, task-related thought, and task-unrelated thought (TUT) reported by each age group, in the episodic specificity induction (ESI) and in the control induction.**

|  |  | ESI | Control |
|---|---|---|---|
| Earworm | Younger | 0.33 (0.64) | 0.25 (0.53) |
|  | Older | None | None |
| External distraction | Younger | 0.13 (0.34) | 0.13 (0.34) |
|  | Older | 0.50 (0.93) | 0.54 (0.72) |
| Task-related | Younger | 1.25 (1.60) | 0.96 (1.12) |
|  | Older | 1.54 (1.69) | 1.92 (1.98) |
| Task-unrelated | Younger | 4.58 (3.23) | 4.75 (3.25) |
|  | Older | 4.29 (3.52) | 4.33 (3.33) |

To characterize whether the lack of finding reflected a true null effect we conducted a Bayesian analysis including the same factors 2 (age group: young, older) × 2 (type of induction: ESI, control). The inverse Bayes factor for age group indicates that the present data is twice as more likely under a null effects model ($BF_{01} = 2.23$), providing weak support for the null hypothesis (following guidelines in [79]). If we consider that the medium effect size of age-related differences in MW in previous studies has been shown to be 0.89 [8] and adapt the prior information accordingly, the inverse Bayes factor for age group ($BF_{01} = 3.20$) indicates moderate support for the null hypothesis. For type of induction ($BF_{01} = 4.55$) the evidence in favor of the null hypothesis was moderate. Thus, these findings suggest that training participants to report episodic details with the ESI does not change the amount of sTUT experienced by participants. The comparison between younger and older adults is in line with the absence of age-related effects found previously. When the size of previous age effects is considered, there is moderate evidence that such difference in the amount of sTUTs experienced by younger and older adults is not observed here.

It should be noted that these analyses are based on the raw number of sTUTs, rather than the proportion of sTUTs out of all spontaneous thoughts. Raw scores, or its proportion by number of probes, are the most common way to measure sTUTs and MW [8], and thus allow us to compare effect sizes in the present study with those of previous studies and, importantly, to use a previous mean effect size based on this type of measure to refine our Bayesian analyses. Further, using the proportions of sTUTs out of all spontaneous thoughts, instead of raw scores, in the 2 (age group: young, older) × 2 (type of induction: ESI, control) mixed ANOVA did not change the results, which continued to show no effect of age group ($p = .317$), type of induction ($p = .583$), or their interaction ($p = .266$).

Finally, the within-participants design allowed us to conduct a correlational analysis between sTUTs frequency in the first and second sessions of the vigilance task. Along with the prediction of no main effect for type of induction, for which we found supporting evidence, we would expect a positive correlation for sTUT frequency within participants. This correlation was indeed positive and significant for both younger ($r = .657$, $p < .001$) and older adults ($r = .818$, $p < .001$), further indicating that the vigilance task reliably elicits sTUTs.

## 3.5. Participant-based classifications

We also investigated the impact of several key variables based on participants' classification on the frequency of sTUTs by including them in separate ANOVAs (see the descriptive statistics for trigger status, temporality and visual/verbal form in the S3 File). To examine the impact of trigger status, we performed a 2 (age group: young, older) × 2 (type of induction: ESI, control) × 2 (trigger status: without trigger, with trigger) mixed ANOVA on the frequency of sTUT. We found a main effect of trigger status, $F(1,46) = 45.22$, $MSE = 475.02$, $p < .001$, $\eta_p^2 = .50$, as sTUTs with a trigger ($M = 3.82$, $SD = 3.10$) were more frequent than sTUTs without a trigger ($M = .68$, $SD = .72$). No other effects or interactions were found ($p$'s $> .67$). The Bayesian analyses were consistent with these results, showing moderate to extreme support for models excluding all effects and interactions ($BF_{Exclusion} > 9.16$) but trigger status ($BF_{Exclusion} < 0.01$). Regarding type of trigger, 76.4% of all spontaneous thoughts were triggered by cue words presented in the task, as in other similar paradigms [36].

To examine the impact of temporality, we performed a 2 (age group: young, older) × 2 (type of induction: ESI, control) × 4 (temporality: past, present, future, atemporal) mixed ANOVA on the frequency of sTUTs. The Greenhouse-Geisser correction was used here and elsewhere to adjust for violations of sphericity. We found a main effect of temporality, $F(2.40,110.36) = 5.25$, $MSE = 22.20$, $p = .004$, $\eta_p^2 = .10$, and post-hoc analyses revealed that past

sTUTs ($M$ = 1.56, $SD$ = 1.66) were more frequent ($p$ = .003) than present sTUTs ($M$ = 0.69, $SD$ = 0.75). Additionally, temporality interacted with age group, $F(2.40,110.36)$ = 3.49, $MSE$ = 14.74, $p$ = .026, $\eta_p^2$ = .07, with post-hoc analyses showing that present sTUTs were more frequent in older than younger adults ($p$ < .001, $M$ = 0.25, $SD$ = 0.74 for YAs and $M$ = 1.13, $SD$ = 0.74 for OAs), with no evidence of age differences in other temporalities ($p \geq$ .07). No other effects or interactions were found ($p$'s > .16). The Bayesian analyses supported models including the temporality effect and the interaction with age group ($BF_{Exclusion}$ < 0.10). For age group the evidence arising from the Bayesian analysis was inconclusive ($BF_{Exclusion}$ = 0.47), while there was strong to extreme support for models excluding type of induction and remaining interactions ($BF_{Exclusion}$ > 20.40).

We examined the impact of the verbal or visual form on the frequency of sTUTs, in a 2 (age group: young, older) × 2 (type of induction: ESI, control) × 2 (form: verbal, visual) mixed ANOVA. This revealed a main effect of form, $F(1,46)$ = 4.54, $MSE$ = 26.25, $p$ = .038, $\eta_p^2$ = .09, which was reflected by less frequent verbal sTUTs ($M$ = 1.85, $SD$ = 1.47) than visual sTUTs ($M$ = 2.59, $SD$ = 2.34). There was also interaction between form and age group, $F(1,46)$ = 7.46, $MSE$ = 43.13, $p$ = .009, $\eta_p^2$ = .14. Post-hoc analyses revealed that verbal sTUTs ($M$ = 1.50, $SD$ = 1.47) were less frequent ($p$ = .001) than visual sTUTs ($M$ = 3.19, $SD$ = 2.34) in younger adults, but frequency did not differ according to form in older adults (verbal: $M$ = 2.21, $SD$ = 1.47; visual: $M$ = 2.00, $SD$ = 2.34). No other effects or interactions were found ($p$'s > .34). This was in line with the Bayesian analysis that showed moderate to extreme support for a model excluding all effects and interactions ($BF_{Exclusion}$ > 7.99) but age group ($BF_{Exclusion}$ = 0.07), form ($BF_{Exclusion}$ = 0.01) and their interaction ($BF_{Exclusion}$ = 0.02).

Finally, we conducted analyses for detail, valence, and arousal on a subsample of 21 YAs and 18 OAs who reported sTUTs in both sessions (ESI and control), in order to assess potential changes on these phenomenological dimensions (see Table 3 for descriptive statistics). We conducted a 2 (age group: young, older) × 2 (type of induction: ESI, control) mixed ANOVA separately for each rating, however, there were no effects or interactions (all $p$'s > .18).

### 3.6. Spontaneous TUTs episodic specificity

**3.6.1 Episodic specificity measures.** We conducted 2 (age group: young, older) × 2 (type of induction: ESI, control) mixed ANOVAs on the overall, strictly episodic scores, and episodic events with and without detail (for a summary these measures, please see Fig 3). There were no significant main effects or interactions (all $p$'s > .10). To ensure that these results were not due to the order in which the experimental manipulation was administered to the participants, we conducted additional analyses and confirmed that the results were not altered by including the order of the type of induction (which was counterbalanced between participants) in the analyses. Additionally, the order of the type of induction did not show significant main effects or interactions ($p$'s $\geq$ .08). To characterize whether the lack of significant findings

**Table 3. Mean ratings (standard deviation) of detail, valence and arousal by age group in the episodic specificity induction (ESI) and in the control induction.**

|  |  | ESI | Control |
|---|---|---|---|
| Detail | Younger | 4.07 (1.26) | 4.27 (0.91) |
|  | Older | 3.56 (1.54) | 3.94 (1.09) |
| Valence | Younger | 3.93 (0.96) | 3.98 (0.53) |
|  | Older | 4.21 (0.86) | 4.14 (0.98) |
| Arousal | Younger | 3.05 (1.39) | 2.85 (1.19) |
|  | Older | 3.25 (1.31) | 3.33 (1.34) |

reflected a true null effect we conducted additional Bayesian analyses. For the overall episodic specificity score, there was weak support for the null or the alternative hypothesis for either the effect of age group ($BF_{01} = 1.80$) and induction ($BF_{01} = 2.13$). However, there was strong evidence in favor of the null ($BF_{01} = 13.21$) for the interaction. If we consider that the effect size of age-related differences in deliberate retrieval for a similar episodic specificity measure in a previous study has been shown to be 1.08 [45], and adapt the prior information accordingly, the inverse Bayes factor for age group ($BF_{01} = 2.99$) indicates moderate support for the null hypothesis. This suggests that episodic specificity in younger and older adults does not differ here as in deliberate retrieval. For the strictly episodic score, there was moderate to strong support for the null hypothesis for both main effects and the interaction ($BF_{01} > 3.88$). A similar result was found for the age effect ($BF_{01} = 5.51$) when adapting the prior based on a previous effect size of 0.74 [45]. For the episodic events with and without detail there was no clear support for age-related differences or their absence ($BF_{01} = 1.03$), but moderate evidence in favor of the null hypothesis for the type of induction and the interaction ($BF_{01} > 4.68$). A similar result is found for the age effect ($BF_{01} = 1.97$) when adapting the prior based on a previous effect size of 1.38 [24]. These findings demonstrate that an induction targeting event construction does not increase the specificity of thoughts retrieved spontaneously. Additionally, older adults do not show a reduction in the number of events and detailed specific events to the degree they do in deliberate retrieval. However, for specific events (with or without detail) the results were inconclusive.

**3.6.2 Episodic specificity categories.** Finally, we investigated the frequency of all types of thoughts as defined by the TEMPau (see Table 4). We conducted a 2 (age group: young, older) × 2 (type of induction: ESI, control) × 5 (TEMPau category: general knowledge, repeated/extended event not situated in time and place, repeated/extended event situated in time and place, specific event situated in time and place without additional details, specific event situated in time and place with additional details) mixed ANOVA. We found a main effect of TEMPau category, $F(2.50, 114.92) = 48.36$, $MSE = 16.57$, $p < .001$, $\eta_p^2 = .27$. Post-hoc analyses revealed that general knowledge sTUTs ($M = 1.77$, $SD = 1.50$) were more frequent than sTUTs in any other category ($p\text{'s} < .001$) except specific events without detail ($p = .09$, $M = 1.04$, $SD = 1.01$). In turn, specific events without detail were more frequent than specific events with detail ($p = .012$, $M = 0.52$, $SD = 1.50$) and repeated/extended events situated in time and place ($p < .001$, $M = 0.32$, $SD = 0.51$). Additionally, the interaction between TEMPau category and type of induction was marginally significant, $F(3.40, 156.47) = 2.51$, $MSE = 2.78$, $p = .054$, $\eta_p^2 = .05$. Post-hoc analyses revealed a reduction ($p = .018$) of general knowledge sTUTs in the ESI

**Table 4. Mean number (standard deviation) of spontaneous task-unrelated thought (TUT) in each TEMPau specificity category, for each age group in the episodic specificity induction (ESI) and in the control induction.**

|  |  | ESI | Control |
|---|---|---|---|
| General knowledge | Younger | 1.29 (1.23) | 1.96 (1.99) |
|  | Older | 1.71 (1.76) | 2.13 (1.68) |
| Repeated/extended event not situated in time and place | Younger | 1.08 (1.72) | 0.71 (1.08) |
|  | Older | 0.92 (1.56) | 0.67 (1.24) |
| Repeated/extended event situated in time and place | Younger | 0.21 (0.51) | 0.33 (0.56) |
|  | Older | 0.50 (0.93) | 0.25 (0.53) |
| Specific event situated in time (<24h) and place without additional details | Younger | 1.29 (1.16) | 1.42 (1.41) |
|  | Older | 0.79 (1.14) | 0.67 (1.20) |
| Specific event situated in time (<24h) and place with additional details | Younger | 0.71 (1.12) | 0.38 (.58) |
|  | Older | 0.38 (0.88) | 0.63 (1.17) |

($M = 1.50$, $SD = 1.52$) compared to the control induction ($M = 2.00$, $SD = 1.84$). The Bayesian analysis showed extreme support of a model including the main effect of TEMPau category ($BF_{Inclusion} > 100$). Additionally, we found moderate and extreme evidence for models excluding all other factors and possible interactions ($BF_{Exclusion} > 7.45$). These results suggest that the interaction found between TEMPau category and type of induction should be taken with caution. In sum, when thoughts were considered in terms of all the TEMPau categories the findings show that thoughts retrieved spontaneously are more frequently either about general knowledge or about non-detailed specific events. Additionally, there was marginally significant reduction in the number of sTUTs describing general knowledge after targeting event construction with the ESI, which was not supported by the Bayesian analysis Importantly, this did not translate in an increase of sTUTs in more specific events categories, for which there were no differences. Finally, the frequency of sTUTs in different TEMPau categories was the same for younger and older adults, supporting the role of spontaneous retrieval in reducing age effects.

**3.6.3. Episodic specificity in past vs future-oriented sTUTs.** To investigate whether the present data replicate previous findings with respect to the similarities and differences between past and future sTUTs, we included temporality as a factor and repeated the analysis by including the overall and strictly episodic score, specific events, and TEMPau category. In order to directly compare past and future, we did not include present and atemporal sTUTs in these analyses. Thus, we conducted separate 2 (age group: young, older) × 2 (type of induction: ESI, control) × 2 (temporality: past, future) mixed ANOVA on each of the measures. First, turning to the overall specificity score, we found a main effect of temporality, $F(1,46) = 8.88$, $MSE = 27.00$, $p = .005$, $\eta_p^2 = .16$, which was due to a greater overall specificity score for past-oriented ($M = 1.41$, $SD = 1.54$) than future-oriented ($M = 0.66$, $SD = 0.84$) sTUTs. Second, for the strictly episodic score we found a similar pattern of results, with a main effect of temporality, $F(1,46) = 13.66$, $MSE = 6.75$, $p = .001$, $\eta_p^2 = .23$, such that past-oriented sTUTs were more specific ($M = 0.43$, $SD = 0.67$) than future-oriented ones ($M = 0.05$, $SD = 0.21$). Third, we found a main effect of temporality on specific events (with and without detail), $F(1,46) = 5.60$, $MSE = 11.02$, $p = .022$, $\eta_p^2 = .11$, with more specific events about the past ($M = 0.97$, $SD = 1.15$) than the future ($M = 0.49$, $SD = 0.72$). There were no other significant main effects or interactions (all $p$'s > .06). The effect closer to significance was a main age group effect for the strictly episodic score $F(1,46) = 3.49$, $MSE = 6.02$, $p = .068$, $\eta_p^2 = .07$, where older adults showed numerically lower specificity ($M = 0.55$, $SD = 0.66$) than younger adults ($M = 0.91$, $SD = 0.66$). We further explored these effects in a Bayesian analyses that showed moderate to extreme evidence for models excluding all variables and interactions but temporality ($BF_{Inclusion} > 100$) for both the overall and strictly episodic scores ($BF_{Exclusion} > 3.88$). For specific events, we found moderate to extreme evidence for models excluding in all variables and interactions ($BF_{Exclusion} > 4.73$) but temporality ($BF_{Inclusion} = 8.88$) and age group ($BF_{Inclusion} = 0.40$). A main effect of temporality described the best model ($BF_{10} = 21.56$), and adding the age group effect decreased the support for the model by a factor of 1.25. Thus, the impact of the age group is inconclusive in this case. In sum, across all three measures of episodic specificity we found more sTUTs related to the past than the future, and no evidence of differences for other variables, including age. For specific events a null effect of age could not be confirmed. Importantly, these findings show that the overall and strictly specificity of past sTUTs is the same in both younger and older adults, and that pattern extends to future sTUTs.

Finally, we included temporality as an additional factor and reexamined the frequency of TEMPau category by conducting a 2 (age group: young, older) × 2 (type of induction: ESI, control) × 5 (TEMPau category: general knowledge, repeated/extended event not situated in time and place, repeated/extended event situated in time and place, specific event situated in

time and place without additional details, specific event situated in time and place with additional details) × 2 (temporality: past, future) mixed ANOVA. We found a main effect of TEMPau category, $F(2.78,128.27) = 10.85$, $MSE = 6.12$, $p < .001$, $\eta_p^2 = .19$. Post-hoc analyses revealed that the main effect of TEMPau category was due to a pattern of differences between categories that was distinct from the overall analysis (including all temporalities). When including past and future sTUTs only, specific events without detail were more frequent than sTUTs in any other category of the TEMPau ($p$'s $\leq .045$). There was also a main effect of temporality, $F(1,46) = 8.13$, $MSE = 5.70$, $p = .007$, $\eta_p^2 = .15$, which was due to a greater frequency of past-oriented sTUTs ($M = 0.31$, $SD = 0.33$) than future sTUTs ($M = 0.16$, $SD = 0.19$). There were no other main effects or interactions (all $p$'s $> .07$). These results were consistent with the Bayesian analyses that showed moderate to extreme evidence for models excluding all variables and interactions ($BF_{Exclusion} > 14.49$), but TEMPau ($BF_{Inclusion} > 100$), temporality ($BF_{Inclusion} = 63.74$) and their interaction ($BF_{Inclusion} = 0.72$). In sum, we found that past sTUTs were more frequent than future sTUTs, and both seem to describe specific events, with no differences between younger and older groups.

## 4. Discussion

The present study examined episodic specificity in descriptions of spontaneous thought in aging. We also tested whether an ESI influenced the nature of information reported during spontaneous retrieval. Overall, we found no effects of age or the ESI, and moderate to extreme evidence for null effects in some of the episodic specificity measures. Our findings suggest that spontaneous retrieval bypasses event constructive processes that support episodic specificity, namely, by providing access to pre-stored event representations [39]. The absent or minimal involvement of event construction during spontaneous retrieval may also contribute to the attenuation of age-related changes in episodic specificity as found here. Below, we discuss these results by exploring the mechanisms supporting episodic specificity in spontaneous thought in aging.

### 4.1. Age-related differences

Replicating our previous study [17], we found no evidence for an age-related decrease in sTUTs frequency when key methodological confounds were controlled for (e.g., involvement of meta-awareness). The absence of age-related differences was supported by moderate evidence for a null age effect, and extended to a more fine-grained analysis of the data that focused only on past-oriented sTUTs (in line with IAM research; see e.g. [23]) and future-oriented sTUTs. For episodic specificity, we also found no age-related differences irrespective of how events were defined. Additionally, there was moderate support for a null effect in the number of events and specific events with detail. When focusing on past and future-oriented sTUTs, we found the same pattern of results. These results demonstrate that there is no consistent age-related decrease in episodic specificity for spontaneous retrieval, replicating previous results using self-report measures [24].

Our findings are in line with the idea that the recall of specific episodic information is supported by different mechanisms depending upon whether retrieval is involuntary/spontaneous or voluntary/deliberate [20]. In particular, it has been suggested that involuntary autobiographical memories and future thoughts emerge through a process based on cue-item discriminability [20], which accesses specific information, bypassing age-related differences in the top-down strategic processes required in deliberate recall. Two additional findings support this interpretation. First, we found that the majority of sTUTs were triggered by a cue, across age-group and type of induction. Second, both past and future sTUTs were more likely to

reflect specific events across age-group. In sum, we found support for the role of spontaneous retrieval in attenuating age-related difficulties to access specific episodic information. In the context of the theories of cognitive aging, the present results extend the empirical support for the key role of reduced cognitive resources in age-related changes [80]. Consistent with this view, we did not find age-related differences in spontaneous retrieval when self-initiated processes are not required (for a review, see reference [1]).

## 4.2. Episodic specificity induction effect

We found no evidence of an ESI effect in either the frequency or episodic specificity of spontaneous thoughts. The ESI did not increase the number of specific events, either with or without detail, as shown by moderate evidence for a null effect in these measures. These findings are in line with a direct and automatic route involved in spontaneous retrieval (for a review, see reference [81]) and support the view that self-initiated processes are the main source of age-related differences in episodic detail (e.g., [82]). We also found the same results when looking at past and future sTUTs separately. During deliberate retrieval, future events have been shown to require more event construction than past events (for a review, see reference [83]). However, here, we found the same pattern of results for both temporal orientations, further supporting the view that spontaneous representations are similarly independent of event construction irrespective of whether they are temporally oriented to the past or future. We found evidence for an ESI effect when considering all TEMPau categories, reflected by a decrease in the amount of general thoughts. However, this decrease did not translate to a significant increase in more specific thoughts. Additionally, the Bayesian analysis did not support this marginally significant result, and thus it remains unclear to what extent the effect is relevant for interpretation. In any case, the evidence shown here suggests that, at least in the case of spontaneous thoughts, the ESI does not increase episodic specificity. We consistently failed to find significant differences in the number of specific events across different measures. It may be, nonetheless, that the ESI has an effect by increasing the amount of detail in thoughts that remain non-specific (not located in time and space). Such an effect would be better captured by scoring of the number of episodic details, which was not possible due to the nature of spontaneous thoughts elicited in the present study. We further discuss how to improve this aspect in the limitations section.

An alternative interpretation for the null ESI effect would be that cue-item discriminability mechanism supports the event construction of episodic details in an automatic fashion. More specifically, event construction would still be required but facilitated and accelerated by a "potent" cue (e.g., [84]), which would constitute a bottom-up constructive route, in addition to the deliberate top-down constructive process [85]. In this case, the spontaneous retrieval process would by itself increase episodic specificity and make the ESI effect redundant. However, if this was the case, there should be a ceiling effect in episodic specificity of spontaneous thoughts. On the contrary, we found only a small number of specific events with detail, indicating that there was room for the ESI effect to influence episodic specificity if spontaneous thoughts rely on event construction.

How can we explain the existence of pre-stored event representations? Mace [81] proposes an explanation for involuntary memory retrieval based on "literal" representations of events, which are conceptually equivalent to pre-stored event representations. He explains these representations in the context of constructive views that admit that "literal" event representation may stem from the episodic memory system [84]. Namely, these would be long-term fragments of event representations that have been previously constructed. However, there has not been, to our knowledge, an experimental test of this idea. Thus, further research is needed to

understand how retrieval of events is possible in the absence of event construction. This is particularly important for spontaneous future events. If there is minimal event construction in spontaneous retrieval, novel future events cannot be spontaneously retrieved. Instead, spontaneous future thoughts would more appropriately be characterized as memories of future thoughts that have been deliberately recalled (and constructed) before. In fact, previous research supports the view that spontaneous future thoughts are "prestored representations of previously imagined events" ([40], p. 269). More recently, the evidence for this idea has been reviewed and developed in the dual process account of future thinking, which suggests that the majority of spontaneous future thoughts are pre-made and result from the activation of representations of previously constructed events [53].

In sum, we found that the ESI does not increase the number of specific events in spontaneous retrieval, consistent with an automatic mechanism that supports episodic specificity in spontaneous thoughts and with the absence or minimal of event construction in spontaneous retrieval.

## 4.3. Phenomenological characteristics

Regarding phenomenological characteristics, we found no age-related differences in self-reported detail, emotional arousal and valence. Thus, we did not replicate previous age-related differences in emotional arousal and valence [21], including an age-related positivity effect in spontaneous thought [24]. However, this result is difficult to interpret due to smaller sample sizes. The analysis of temporality in all sTUTs revealed there were more past than present-oriented thoughts. Present sTUTs were, in turn, more frequent for older adults, but the Bayesian analysis did not support this effect. When comparing only past and future-oriented sTUTs, the former were more frequent across age-group and type of induction. These results are in line with similar studies that report more past than present [86] and future-oriented sTUTs [62]. Finally, younger but not older adults showed more visual than verbal imagery, consistent with an age-related decrease in visual imagery ability [87]. Visual imagery is closely related to episodic specificity (e.g., [88]). Therefore, how can we explain that there are age-related differences in visual imagery but not in episodic specificity? This may be due to younger adults more frequently reporting visual images without any episodic context or event associated (e.g., "When I saw the word "rotten", I saw a rotten apple in my mind. There was not a specific context or time").

## 4.4. Limitations and future directions

One limitation of the current study is that we did not analyze episodic specificity based on internal and external details using the AI procedure [26], because the spontaneous thoughts examined contained more implicit information and were shorter descriptions. Although we distinguished between specific events without versus with episodic detail, it was not possible to quantify how much more detail there was, which reduced the precision of the analyses. To capture more explicit aspects of spontaneous thoughts, one alternative would be to include a post-recall deliberate elaboration after both spontaneous and deliberate retrieval. If spontaneous retrieval supports the automatic access to episodic detail then it should also facilitate subsequent deliberate elaboration of these same events and attenuate age-related differences compared to deliberate retrieval. Our findings highlight that it will be important for future research to consider the distinction between a specific event, as defined categorically with the TEMPau, and episodic detail as a continuum. Although it seems logical to assume that increasing episodic detail leads to more time and place details, and thus to more specific events, some studies have shown a dissociation between detail and specificity that is thought to reflect dissociations in underlying processes [89].

In the future, it will also be important to directly compare spontaneous with deliberate thoughts using the same experimental paradigm (e.g., [90]). Given the consistency of age-related decrease in episodic specificity in a variety of tasks (e.g., [10,24,26,28,31,34,45], it is likely that we would have found a similar effect here if we had asked older adults to generate deliberate thoughts. However, a direct comparison between deliberate and spontaneous conditions will provide conclusive evidence.

In our analyses there was an unequal number of sTUTs *per* participant, due to the unexpected nature of sTUTs and the consistent individual differences in sTUTs (e.g., [91]). This contrasts with previous ESI studies on deliberate memory and future thinking [11], and may have played a role in the absence of the ESI effect. Additionally, it excludes participants that do not report sTUTs in both sessions from the analyses on phenomenological characteristics. To equate sTUTs between subjects, future studies may use experience-sampling methods that probe participants until a certain number of sTUTs are recorded. Similar methods have been successful in studying spontaneous thoughts with both younger and older adults [18].

Finally, it is also worth noting that although education and gender were similarly distributed in the younger and older group, both included mainly women participants. Although we are not aware of any evidence indicating that an ESI effect would interact with gender, future studies including more men participants are important to dismiss the possibility of a gender related bias.

## 5. Conclusions

We found that age-related differences in episodic specificity are attenuated in spontaneous retrieval. Additionally, training participants to recall episodic detail did not increase episodic specificity in subsequent spontaneous thought. These findings are consistent with the view that episodic specificity in spontaneous thought is supported by automatic cue-related mechanisms that bypass event construction. Several questions remain to be further explored in paradigms that include comparisons with deliberate and directly retrieved thoughts and that allow participants to elaborate their spontaneous thoughts. Nonetheless, the present evidence shows that activating spontaneous retrieval, for example by using environments embedded with personal specific cues (see [92] for an empirical demonstration with Alzheimer's disease patients), is a promising strategy to support episodic specificity in old age.

## Supporting information

**S1 File. Vigilance task—words and schematic representation.**
(DOCX)

**S2 File. Coding instructions.**
(DOCX)

**S3 File. Descriptive statistics of spontaneous task-unrelated thoughts by trigger, temporality and verbal/visual form.**
(DOCX)

## Acknowledgments

The authors thank to Dr. Kevin Madore for providing access to the episodic specificity induction materials, and to Ana Carolina Artiaga, Bianca Gerardo, and Kevin Ramos for their support in the coding process.

## Author Contributions

**Conceptualization:** Magda Jordão, Maria Salomé Pinho, Peggy L. St. Jacques.

**Data curation:** Magda Jordão.

**Formal analysis:** Magda Jordão, Maria Salomé Pinho, Peggy L. St. Jacques.

**Funding acquisition:** Magda Jordão.

**Investigation:** Magda Jordão.

**Methodology:** Magda Jordão, Maria Salomé Pinho, Peggy L. St. Jacques.

**Project administration:** Magda Jordão.

**Supervision:** Maria Salomé Pinho, Peggy L. St. Jacques.

**Validation:** Maria Salomé Pinho, Peggy L. St. Jacques.

**Visualization:** Magda Jordão.

**Writing – original draft:** Magda Jordão, Maria Salomé Pinho, Peggy L. St. Jacques.

**Writing – review & editing:** Magda Jordão, Maria Salomé Pinho, Peggy L. St. Jacques.

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
