## [Decision Letter · Decision Letter 0]

6 May 2020

PONE-D-20-07424

The effects of aging and an episodic specificity induction on spontaneous task-unrelated thought

PLOS ONE

Dear Ms Jordao,

Thank you for submitting your manuscript to PLOS ONE. After careful consideration, we feel that it has merit but does not fully meet PLOS ONE’s publication criteria as it currently stands. Therefore, we invite you to submit a revised version of the manuscript that addresses the points raised during the review process.

As you can see in the reviewer comments, both reviewers positively evaluate your work. However, both ask for more details and clarification, in particular with regard to methods, procedures, and variables (mainly in the Methods and Results sections). These points are essential to meeting PLOS ONE's publication criteria. The reviewers also provide excellent suggestions for further connecting your work to previous research and current theories, and offer interesting suggestions for follow-up analyses that might provide insight into the generalizability and stability of the findings. 

We would appreciate receiving your revised manuscript by Jun 20 2020 11:59PM. To enhance the reproducibility of your results, we recommend that if applicable you deposit your laboratory protocols in protocols.io, where a protocol can be assigned its own identifier (DOI) such that it can be cited independently in the future. For instructions see: http://journals.plos.org/plosone/s/submission-guidelines#loc-laboratory-protocols

We look forward to receiving your revised manuscript.

Kind regards,

Myrthe Faber

Academic Editor

PLOS ONE

Reviewers' comments:

Reviewer's Responses to Questions

**Comments to the Author**

1. Is the manuscript technically sound, and do the data support the conclusions?

Reviewer #1: Yes

Reviewer #2: Yes

2. Has the statistical analysis been performed appropriately and rigorously? 

Reviewer #1: Yes

Reviewer #2: Yes

3. Have the authors made all data underlying the findings in their manuscript fully available?

Reviewer #1: No

Reviewer #2: Yes

4. Is the manuscript presented in an intelligible fashion and written in standard English?

Reviewer #1: Yes

Reviewer #2: Yes

5. Review Comments to the Author

Reviewer #1: Summary

This study investigated age-related differences in episodic specificity of spontaneous thoughts. Thoughts were collected during a vigilance task on two occasions, once following an episodic specificity induction and once after a control induction. Thoughts characteristics were partly self-evaluated (e.g. spontaneous/deliberate) and partly coded by experimenters (e.g. level of episodic specificity). Results present no age-related, nor induction-related differences on spontaneous thoughts’ episodic specificity.

Review

This is an interesting paper which helps further understand spontaneous thoughts in ageing by applying methodologies typically used for deliberate retrieval of memories or imagined events. The paper is well written and provides a complete theoretical background as well as an interesting methodological approach.

Minor point

298-368: The main recommendation is a need for more clarity in the description of variables measured. Quite a large number of characteristics are measured based on participants report. Each measure is described in detail, however, because of their number and strong similarity for some, it would be important to add a small paragraph restating briefly the name and purpose of these dependent variables. A graph may be considered to help visualise the process by which these measures were computed. This should enable a better understanding of the study and the later outlined results.

Other recommendations are as follow:

203: A threshold for BDI scores needs to be provided. Additionally, it is not mentioned whether some participants have been excluded based on exclusion criteria. Presence or absence of any compensation for participants’ time and travel should be stated.

253: From reading the paper, it seems that information about how motivation was evaluated is missing. In the same line (283), it is unclear whether questions were asked verbally by the experimenter or presented on the computer screen.

438: A word seems to be missing in this sentence.

506: potentially related to the lack of clarity in the method section. It is difficult to see what is referred to by “the overall, strictly episodic scores, and episodic events with and without detail”.

540-454: This section needs to be reworded, potentially cutting the sentence, to help the reader fully capture key information.

565: Problems with table format.

567-594: In this part of the results section, analyses are conducted on overall specificity scores. However, results are reported only in terms of frequency (e.g. “past-oriented sTUTs were more frequent (M = 0.43, SD = 0.67) than future-oriented ones (M = 0.05, SD = 0.21)”). This is confusing, this section is supposed to evaluate episodic specificity in past and future thoughts, yet results mention that past thoughts are more frequent then future one without mention of episodic specificity. It is probably just a case of rewording to enable better understanding.

582: It is stated that all other results are p >.06. For completeness, authors may consider stating which effect is, to some extent, approaching significance. This is particularly important as elsewhere in the manuscript a marginal effect has been reported (p<.054). The information provided so far may bring questions regarding selection of information. To be more transparent and consistent, p-values close to .06 should be reported with more details.

SM2: It is unclear to which coding aspect the “Note” section is referring too.

Reviewer #2: The paper reports a new study which investigated the effects of age on the number of spontaneous task unrelated thoughts (sTUTs) using a modified vigilance task in which participants were randomly stopped 12 times to report what were they thinking about just before being stopped. Unlike more standard tasks (such as SART in mind wandering research) participants were exposed to incidental word cues that could trigger sTUTs in participants. The main (and novel) aim of the study was to examine the specificity of sTUTs as a function of age group (young versus old) and the episodic specificity induction (ESI) manipulation (ESI versus control). The key finding was that there were no age effects in the number of sTUTs and various measures of specificity of reported thoughts.

The study is carefully designed, well conducted and reports new findings that should be of interest to researchers of cognitive ageing, mind-wandering, episodic future thinking and involuntary autobiographical memory. The analyses of variance are followed up by Bayesian analyses. The conclusions based on these analyses are justified and appropriate, even if they contradict some of the findings that have been reported in mind wandering research and recently reviewed by Jordão et al. (2019) and Maillet and Schacter (2016). However, the findings seem to be in line with a growing body of empirical studies which demonstrate absence of age effects when using more naturalistic tasks and situations in which participants rely on more spontaneous retrieval processes than on more top down strategic effortful retrieval processes (e.g., Berntsen, Rasmussen et al., 2017, Psychology & Aging; Jordão et al., 2019, Psychological Research; Kvavilashvili, Mirani, et al., 2010; Psychology & Aging; Qin et al., 2014, PLoS One; Warden et al., 2019, Psychological Research). Although ideally I would have liked to see larger samples in this study, the findings are fairly informative and interesting to merit publication subject to some revisions as specified below. 2. Although the paper is well written overall, the revisions should primarily concentrate on the method and results sections to increase the clarity. Indeed, I felt that several important methodological details were not explained with sufficient clarity.

MAJOR POINTS

The within subjects design allows authors to examine one very interesting question that was not addressed in the paper. Namely, it would be very interesting to see if the number of sTUTs correlates positively across the two sessions. This should be possible to do given that there was no significant effect of ESI. I realise that this is perhaps a minor issue for the authors, but I was wondering if it was possible to check this and report perhaps somewhere in the results or discussion section?

Method section

1. Lines 243-245 – Please provide more details about the episodic specificity induction procedure which will be useful to readers who are unfamiliar to it. For example, what are the instructions that participants receive? What exactly did the videos contain, are these videos comparable?

2. Lines 253-270 – Section 2.5 describing the vigilance task was very difficult to understand. In fact I could not understand at all what were the target stimuli, how did the words appear on trials, what words were they, etc etc? Many sentences are unclear. For example, “ Prior to starting the task participants assessed their motivation level, to avoid the influence of perceived performance on these ratings (58)” (Can you please explain what did this involve?). OR “Our earlier study found no evidence of age differences in MW frequency (17), suggesting that the adaptation controlled for confounding factors, such as possible age-related differences in the use of response options (8)” (what does this mean? Can you explain what response option means here?). OR “Second, we controlled for the number and characteristics of the word cues presented between each probe. Third, we changed the ISI from 7 to 3 seconds, which was important because longer ISIs require spontaneous thoughts triggered by word-cues to be maintained longer to be caught in the probes. Thus, by reducing the ISI, we are able to record both longer and shorter thoughts”. OR “The final task took 15.9 minutes, and randomly presented 72 words and 87 five-point sequences for 3 seconds, followed by a 3 seconds ISI” .

It is unclear in these descriptions how many trials were used, what were target and non-target stimuli, how many trails had words and how many trials did not have any words on them, how were the words selected, did they differ in valence, etc. Did the vigilance task last for 15.9 minutes without the thought probes or with thought probes? How many trials were there between two consecutive probes? What do fife-point sequences refer to? Etc.

3. In section 2.6.1, please provide examples of task-unrelated and task-related thoughts provided by participants. On line 314-315 can you please provide some examples of external distracters, as the definition provided is a bit unclear and long-winded?

4. In section 2.6.3 – please indicate who were the coders? Any of the authors or researchers helping to conduct the study?

Results

1. Section 3.3 – Can you please start this section by providing some descriptive information about a total number of thought probes obtained in 24 young and 24 old participants in ESI and Control conditions (i.e., 288 probes in each of the 4 conditions), an how many probes were “no thoughts”, “earworms”, “distractions” etc? This could then be followed up by means presented in Table 2. However, given that there were probably very few instances of earworms (as few as “no thoughts”) is it worth presenting earworm category in Table 2? It will be also useful to have some descriptive information on reports of thought triggers and if they were predominantly the cue words presented in trials preceding the thought probe?

2. Lines 471-472 - It would be clearer if you stated that participants had significantly higher number of past than present thoughts, and it would be also useful to report comparisons and means for future thoughts.

3. Section 3.6.3 – The 2 (age group: young, older) × 2 (type of induction: ESI, control) × 2 (temporality: past, future) mixed ANOVA on each of the specificity measures may potentially involve a loss of number of participants who did not report either past or future sTUTs. Can you please confirm that this was not the case? I.e., did all participants report at least one past and one future thought?

General discussion

1. Lines 697-700 – It is pointed out that “Instead, spontaneous future thoughts would more appropriately be characterized as memories of future thoughts that have been deliberately recalled (and constructed) before. In fact, previous research supports the view that spontaneous future thoughts are “prestored representations of previously imagined events” (38, p. 269). In relation to this idea, it would be very highly relevant to mention or discuss the dual-processes account of future thinking proposed by Cole and Kvavilashvili (2020) which develops this idea into a coherent theoretical approach and reviews supporting empirical evidence.

Minor Points

1. Please use past tense in the abstract

2. Line 62 – Please define what is episodic specificity immediately after using this term.

3. Line 81 – you could mention here two naturalistic studies by Warden et al. who consistently showed no age effects on involuntary past and future thoughts using diary and experience sampling methods? A study by Gardner and Ascoli (2015; Psychology and Aging) is also highly relevant.

4. Lines 159-160 – It is pointed out that “Alternatively, it has been suggested that direct retrieval processes that characterize spontaneous thought imply the existence of pre-stored event representations, independent of event construction both for autobiographical recall and for episodic future thinking”. This is an important statement and it might be useful to slightly expand on this idea by citing studies of Jeunehomme and D’Argembeau (2016) and Cole and Kvavilahsvili (2020, in press in Psychological Research)?

5. Lines 208- provide age ranges (what was the minimum and maximum age) in each group

6. Lines 214-216 in Design – what were your main DVs?

7. Line 220- Should be “Both sessions included an initial induction procedure (episodic or control), in which a video was presented, followed by a vigilance task to elicit spontaneous thoughts”.

8. Line 224 – Should be “There was an exception”

9. Lines 445 and 449 – should be “sTUTs out of all spontaneous thoughts”

10. Line 506 – it would be helpful to refer readers to Fig 2 and define in brackets what each category refers to (i.e., scores >1, scores = 4, and scores 3 and 4, respectively)

11. Line 528 – “that inducing a targeting event construction does not” – this may need some re-wording?

12. Lines 555 and 668 – I think referring to marginal findings as “anecdotal” is incorrect. You need to refer to them as “marginally significant”

13. Line 650 – should be “find age-related differences”

Figure 1 – instead of Elicitation task may be you should say Vigilance task?

6. PLOS authors have the option to publish the peer review history of their article (what does this mean?). If published, this will include your full peer review and any attached files.

Reviewer #1: No

Reviewer #2: No

---

## [Author Response · Author response to Decision Letter 0]

1 Jun 2020

Dear Dr. Farber,

We are submitting the revised manuscript PONE-D-20-07424, “The effects of aging and an episodic specificity induction on spontaneous task-unrelated thought”, for your consideration. We were pleased with the overall positive reviews of our manuscript and would like to thank the editor and reviewers for the helpful comments. 

In the present manuscript, we introduced changes to clarify our methods, results and variables, following each of the queries raised by the reviewers. To further clarify our methods, we created a new figure (Fig 3) to summarize the coding process and dependent variables extracted. We are also submitting a new version of Fig 2 in which we changed the reference number according with the new references list.

We noticed that one of the reviewers answered no to the question “3. Have the authors made all data underlying the findings in their manuscript fully available?” We would like to enquire if this has to do with the questions about our results that we now clarify in the reply, or if there was any problem with our database content or the link we provided to access the database (https://osf.io/cn28e/?view_only=ef7d717be39b4f91abbddf59abb56c4b). If there is any difficulty accessing the relevant information, we’ll be happy to correct the situation.

Additionally, we found the suggestions on how to better connect our study with previous research and on follow-up analyses very valuable, and introduced them in our manuscript. Please see the “Response to Reviewers”, below, for the full, point-by-point, description of our changes and clarifications. We hope you will find our modifications appropriate. 

Sincerely,

Magda Jordão

*

Reviewer #1

Please note that the indications for lines in our replies refer to the manuscript with track changes

Minor point

298-368: The main recommendation is a need for more clarity in the description of variables measured. Quite a large number of characteristics are measured based on participants report. Each measure is described in detail, however, because of their number and strong similarity for some, it would be important to add a small paragraph restating briefly the name and purpose of these dependent variables. A graph may be considered to help visualise the process by which these measures were computed. This should enable a better understanding of the study and the later outlined results.

R: Thank you for this suggestion. We created a new figure to summarize and clarify the coding and resulting variables (Fig. 3), and added a paragraph that briefly restates the variables at the end of this section (lines 402-407).

Other recommendations are as follow:

203: A threshold for BDI scores needs to be provided. 

R: We’ve now added the BDI threshold on lines 212-213.

Additionally, it is not mentioned whether some participants have been excluded based on exclusion criteria. 

R: Thank you for noticing this. We’ve now added the information about the number of participants who were excluded based on the exclusion criteria, on lines 216-219. 

Presence or absence of any compensation for participants’ time and travel should be stated.

R: This is also now added on lines 197-199.

253: From reading the paper, it seems that information about how motivation was evaluated is missing.

R: Thank you for highlighting this issue. We’ve now added information about the scale we used to measure motivation, on lines 276-277. 

In the same line (283), it is unclear whether questions were asked verbally by the experimenter or presented on the computer screen.

R: We know clarify that questions were presented verbally by the experimenter on line 312.

438: A word seems to be missing in this sentence.

R: We’ve revised the sentence and added the missing word, on line 490. 

506: potentially related to the lack of clarity in the method section. It is difficult to see what is referred to by “the overall, strictly episodic scores, and episodic events with and without detail”.

R: Following your suggestion for the methods, we’ve now added a figure (Fig 3) that summarizes what these measures represent and to make this clearer. We now refer back to the Fig 3 here (lines 566-567).

540-454: This section needs to be reworded, potentially cutting the sentence, to help the reader fully capture key information.

R: I believe the reviewer means 540-545. We’ve reworded this sentence now, cutting the sentence, as suggested (lines 601-605).

565: Problems with table format.

R: Thank you for letting us now you had problems with the table. We’ve now rearranged the table to better fit the manuscript page (line 627). For consistency in the layout of the variables (namely, age group and type of induction), we’ve also rearranged the other tables. 

567-594: In this part of the results section, analyses are conducted on overall specificity scores. However, results are reported only in terms of frequency (e.g. “past-oriented sTUTs were more frequent (M = 0.43, SD = 0.67) than future-oriented ones (M = 0.05, SD = 0.21)”). This is confusing, this section is supposed to evaluate episodic specificity in past and future thoughts, yet results mention that past thoughts are more frequent then future one without mention of episodic specificity. It is probably just a case of rewording to enable better understanding.

R: Thank you for pointing this out. We now reword this section to improve clarity (section 3.6.3., starting on line 629).

582: It is stated that all other results are p >.06. For completeness, authors may consider stating which effect is, to some extent, approaching significance. This is particularly important as elsewhere in the manuscript a marginal effect has been reported (p<.054). The information provided so far may bring questions regarding selection of information. To be more transparent and consistent, p-values close to .06 should be reported with more details.

R: We now provide more information about the effect that is approaching significance on lines 645-648.

SM2: It is unclear to which coding aspect the “Note” section is referring too.

R: Thank you for highlighting this issue. In a revised version of the SM2, we now specify that the “Note” section refers to the episodic specificity coding.

Reviewer #2

Please note that the indications for lines in our replies refer to the manuscript with track changes

MAJOR POINTS

The within subjects design allows authors to examine one very interesting question that was not addressed in the paper. Namely, it would be very interesting to see if the number of sTUTs correlates positively across the two sessions. This should be possible to do given that there was no significant effect of ESI. I realise that this is perhaps a minor issue for the authors, but I was wondering if it was possible to check this and report perhaps somewhere in the results or discussion section?

R: Thank you for this interesting suggestion. We now include this analysis, that turned out to be a significant positive correlation, on lines 496-501.

Method section

1. Lines 243-245 – Please provide more details about the episodic specificity induction procedure which will be useful to readers who are unfamiliar to it. For example, what are the instructions that participants receive? What exactly did the videos contain, are these videos comparable?

R: We’ve now added some more details about the instructions and the content of the videos, on lines 262-264.

2. Lines 253-270 – Section 2.5 describing the vigilance task was very difficult to understand. In fact I could not understand at all what were the target stimuli, how did the words appear on trials, what words were they, etc etc? Many sentences are unclear. For example, “ Prior to starting the task participants assessed their motivation level, to avoid the influence of perceived performance on these ratings (58)” (Can you please explain what did this involve?). OR “Our earlier study found no evidence of age differences in MW frequency (17), suggesting that the adaptation controlled for confounding factors, such as possible age-related differences in the use of response options (8)” (what does this mean? Can you explain what response option means here?). OR “Second, we controlled for the number and characteristics of the word cues presented between each probe. Third, we changed the ISI from 7 to 3 seconds, which was important because longer ISIs require spontaneous thoughts triggered by word-cues to be maintained longer to be caught in the probes. Thus, by reducing the ISI, we are able to record both longer and shorter thoughts”. OR “The final task took 15.9 minutes, and randomly presented 72 words and 87 five-point sequences for 3 seconds, followed by a 3 seconds ISI” . 

It is unclear in these descriptions how many trials were used, what were target and non-target stimuli, how many trails had words and how many trials did not have any words on them, how were the words selected, did they differ in valence, etc. Did the vigilance task last for 15.9 minutes without the thought probes or with thought probes? How many trials were there between two consecutive probes? What do fife-point sequences refer to? Etc.

R: Thank you for highlighting these issues. We’ve now tried to clarify our task following your suggestions. Below we address each on of the point raised about the vigilance task description.

“ Prior to starting the task participants assessed their motivation level, to avoid the influence of perceived performance on these ratings (58)” (Can you please explain what did this involve?).

R: We’ve now added information about the motivation level assessment and rephrased its purpose, on lines 275-278.

OR “Our earlier study found no evidence of age differences in MW frequency (17), suggesting that the adaptation controlled for confounding factors, such as possible age-related 

differences in the use of response options (8)” (what does this mean? Can you explain what response option means here?)

R: We’ve now clarified that response options refer to pre-determined response options, in a forced-choice format, used to categorize what was on the participant mind (as now added to the manuscript on lines 286-287).

OR “Second, we controlled for the number and characteristics of the word cues presented between each probe. Third, we changed the ISI from 7 to 3 seconds, which was important because longer ISIs require spontaneous thoughts triggered by word-cues to be maintained longer to be caught in the probes. Thus, by reducing the ISI, we are able to record both longer and shorter thoughts”.

R: We’ve now rephrased for clarity the controls that have been put in place between each probe and the rational for using a 3 seconds ISI. For the ISI we dropped the comparison with the previous task and instead tried to clarify how we thought it could benefit the present task (on lines 289-297).

OR “The final task took 15.9 minutes, and randomly presented 72 words and 87 five-point sequences for 3 seconds, followed by a 3 seconds ISI” . It is unclear in these descriptions how many trials were used

R: We’ve now clarified that the vigilance task included 159 trials (line 300).

… what were target and non-target stimuli,

R: We’ve now clarified that the stimuli presented in the color yellow were the target (line 303). 

…how many trails had words and how many trials did not have any words on them

R: We’ve now clarified that we either presented words OR five-point sequences, so there were 72 words presented (lines 298-300). We also rephrased five-point sequences to five-dot sequences, which hopefully will be clearer, and added in brackets that we mean this stimulus . . . . . Finally, we referred to the visual representation of the task in the Supplementary Material 1.

…how were the words selected, did they differ in valence, etc. 

R: The words were selected from a well-known word database (as now mentioned on lines 279-280) that provided data on valence, arousal, frequency, concreteness and imageability. We used these data to match the words selected for each version of the vigilance task. The words used and the statistical comparison between versions for each word characteristic is presented in the Supplementary Material 1, as we now mention in the text (line 281).

Did the vigilance task last for 15.9 minutes without the thought probes or with thought probes? 

R: 15.9 minutes was the presentation time for the task, without the probes, as now added on line 298.

How many trials were there between two consecutive probes? 

R: This was variable, in order to avoid making the thought probe predictable. There were 9, 13 or 18 trials between probes, as now added to the manuscript on line 305.

What do fife-point sequences refer to? Etc.

R: As mentioned above, these were sequences of five dots . . . . . that were presented interchangeably with the words. Please check lines 299-300, where we rephrased the expression from five-point sequences to five-dot sequences, and also the Supplementary Material 1 (S1 File), where this is shown in a schematic figure of the vigilance task.

3. In section 2.6.1, please provide examples of task-unrelated and task-related thoughts provided by participants. On line 314-315 can you please provide some examples of external distracters, as the definition provided is a bit unclear and long-winded?

R: We know present examples for task-related thoughts (on lines 339-340) and external distractions (on lines 346-347). We also rephrased the definition of external distractions (on lines 343-346). For task-unrelated thoughts we added a note to the Fig 2, which includes several examples of sTUTs (with different levels of specificity), on lines 340-341. 

4. In section 2.6.3 – please indicate who were the coders? Any of the authors or researchers helping to conduct the study?

R: Thank you for your question. Several students at an undergraduate and graduate level collaborated in this research, as an optional credit or as part of their research assistant role duties. MJ was also a coder and that is also now clearly identified, along with a correction on the information about blinding – MJ was naturally not blind to the hypothesis. Thank you for highlighting this aspect and allowing us to make this correction. You can find this info now summarized on section 2.6.3, lines 414-421.

Results

1. Section 3.3 – Can you please start this section by providing some descriptive information about a total number of thought probes obtained in 24 young and 24 old participants in ESI and Control conditions (i.e., 288 probes in each of the 4 conditions), an how many probes were “no thoughts”, “earworms”, “distractions” etc? This could then be followed up by means presented in Table 2. However, given that there were probably very few instances of earworms (as few as “no thoughts”) is it worth presenting earworm category in Table 2? It will be also useful to have some descriptive information on reports of thought triggers and if they were predominantly the cue words presented in trials preceding the thought probe?

R: We’ve now added the number of overall probes (on line 455), and the number of probes in which spontaneous thought was reported by age group and type of thought (lines 458-461).

Regarding the presentation of the data about earworms, we believe it is important to include this information because we found a different pattern of age-related differences (younger adults experience more earworms). This is in line with previous research that we mention on lines 349-350 and suggests that “musical memories rely on a different cognitive system and show distinct age-related patterns”. 

We’ve also added information about type of trigger, confirming they were predominantly the cue words presented in the task previously to the probe (lines 523-534).

2. Lines 471-472 – It would be clearer if you stated that participants had significantly higher number of past than present thoughts, and it would be also useful to report comparisons and means for future thoughts.

R: We rephrased the manuscript according to the suggestion (line 530). Here and throughout the manuscript we only detailed significant post-hoc comparison results, for the sake of simplicity. However, the detailed data for all temporal orientations, including means and SDs per group, per condition and overall, is presented in the Supplementary Material 3 (S3 File), as mentioned in the beginning of this section (lines 514-515).

3. Section 3.6.3 – The 2 (age group: young, older) × 2 (type of induction: ESI, control) × 2 (temporality: past, future) mixed ANOVA on each of the specificity measures may potentially involve a loss of number of participants who did not report either past or future sTUTs. Can you please confirm that this was not the case? I.e., did all participants report at least one past and one future thought?

R: Not all participants reported at least one past and one future thought. However, we did not exclude these participants from the analysis of specificity, because the specificity scores are based on the raw number of thoughts (of different levels of specificity, as now represented in Fig 3). Being so, when the participants did not have a past or future sTUT in one of the conditions, their score is simply zero, which is still a valid data point, and this participants’ data still feeds into the analysis. For example, if an older participant reports only past sTUTs, this will inform the analysis of the interaction between age group and temporality. Namely, this helps us to analyze if the age-related pattern of specificity in past-oriented thoughts is similar to the one found in the overall analysis. 

We think it is also important to consider the disadvantages of excluding these participants. Unless we have reasons to believe that their data is for some reason unreliable, which we didn’t in our study, the fact that some participants do not report at least on past and future sTUT is just part of the natural inter-variability of this phenomenon. It would seem that not including them in the analysis would pose the risk of biasing our results and interpretations. Thus, it seems to us that the best option is to use a type of analysis that makes use of as much information as possible, which we try to by using frequencies as detailed above. 

General discussion

1. Lines 697-700 – It is pointed out that “Instead, spontaneous future thoughts would more appropriately be characterized as memories of future thoughts that have been deliberately recalled (and constructed) before. In fact, previous research supports the view that spontaneous future thoughts are “prestored representations of previously imagined events” (38, p. 269). In relation to this idea, it would be very highly relevant to mention or discuss the dual-processes account of future thinking proposed by Cole and Kvavilashvili (2020) which develops this idea into a coherent theoretical approach and reviews supporting empirical evidence.

R: Thank you for mentioning this relevant reference. We’ve now added it to the manuscript (lines 766-768).

Minor Points

1. Please use past tense in the abstract

R: We’ve adapted the abstract accordingly (line 28).

2. Line 62 – Please define what is episodic specificity immediately after using this term.

R: We’ve now added the definition on lines 62-64. 

3. Line 81 – you could mention here two naturalistic studies by Warden et al. who consistently showed no age effects on involuntary past and future thoughts using diary and experience sampling methods? A study by Gardner and Ascoli (2015; Psychology and Aging) is also highly relevant.

R: Thank you for mentioning these relevant references. We’ve now added these to the manuscript (lines 83-84).

4. Lines 159-160 – It is pointed out that “Alternatively, it has been suggested that direct retrieval processes that characterize spontaneous thought imply the existence of pre-stored event representations, independent of event construction both for autobiographical recall and for episodic future thinking”. This is an important statement and it might be useful to slightly expand on this idea by citing studies of Jeunehomme and D’Argembeau (2016) and Cole and Kvavilahsvili (2020, in press in Psychological Research)?

R: We’ve now expanded this idea including the references suggested (line 163-168).

5. Lines 208- provide age ranges (what was the minimum and maximum age) in each group

R: We’ve now added this information to the manuscript (lines 220-221).

6. Lines 214-216 in Design – what were your main DVs?

R: We’ve now added this information to the manuscript (lines 230-233).

7. Line 220- Should be “Both sessions included an initial induction procedure (episodic or control), in which a video was presented, followed by a vigilance task to elicit spontaneous thoughts”.

R: Thank you for the correction, now added to the manuscript (lines 237-238).

8. Line 224 – Should be “There was an exception”

R: Thank you for the correction, now added to the manuscript (line 241).

9. Lines 445 and 449 – should be “sTUTs out of all spontaneous thoughts”

R: Thank you for the correction, now added to the manuscript (lines 503 and 507).

10. Line 506 – it would be helpful to refer readers to Fig 2 and define in brackets what each category refers to (i.e., scores >1, scores = 4, and scores 3 and 4, respectively)

R: Following a suggestion of Reviewer 1, we’ve now created a new figure (Fig 3) that represents which events are included in each episodic measure. We’re referring to the Fig 3 as a reminder of what each of the measures mean (lines 566-567).

11. Line 528 – “that inducing a targeting event construction does not” – this may need some re-wording?

R: Yes, this should be “an induction targeting event construction”, thank you for pointing this out (now corrected on line 589).

12. Lines 555 and 668 – I think referring to marginal findings as “anecdotal” is incorrect. You need to refer to them as “marginally significant”

R: Thank you, this is now corrected (lines 616 and 733-734).

13. Line 650 – should be “find age-related differences”

R: Thank you, this is now corrected (lines 716).

Figure 1 – instead of Elicitation task may be you should say Vigilance task?

R: Thank you, this is now corrected in the new version of Fig 1.

---

## [Decision Letter · Decision Letter 1]

16 Jul 2020

PONE-D-20-07424R1

The effects of aging and an episodic specificity induction on spontaneous task-unrelated thought

PLOS ONE

Dear Dr. Jordao,

Thank you for submitting your manuscript to PLOS ONE. After careful consideration, we feel that it has merit but does not fully meet PLOS ONE’s publication criteria as it currently stands. Therefore, we invite you to submit a revised version of the manuscript that addresses the points raised during the review process.

As you can see below, both reviewers positively evaluated your revision. Before your manuscript can be accepted, there are still a few points pertaining to the methods and analyses that need to be clarified (raised by Reviewer 2). I would like to invite a revision that addresses these points.

We look forward to receiving your revised manuscript.

Kind regards,

Myrthe Faber

Academic Editor

PLOS ONE

Reviewers' comments:

Reviewer's Responses to Questions

**Comments to the Author**

1. If the authors have adequately addressed your comments raised in a previous round of review and you feel that this manuscript is now acceptable for publication, you may indicate that here to bypass the “Comments to the Author” section, enter your conflict of interest statement in the “Confidential to Editor” section, and submit your "Accept" recommendation.

Reviewer #1: All comments have been addressed

Reviewer #2: (No Response)

2. Is the manuscript technically sound, and do the data support the conclusions?

Reviewer #1: Yes

Reviewer #2: Yes

3. Has the statistical analysis been performed appropriately and rigorously? 

Reviewer #1: Yes

Reviewer #2: Yes

4. Have the authors made all data underlying the findings in their manuscript fully available?

Reviewer #1: Yes

Reviewer #2: Yes

5. Is the manuscript presented in an intelligible fashion and written in standard English?

Reviewer #1: Yes

Reviewer #2: Yes

6. Review Comments to the Author

Reviewer #1: (No Response)

Reviewer #2: Overall, the revision reads very well and the authors have tried to address most of my comments and suggestions, except the one on the vigilance task. Despite the additions made, the description of the vigilance task in Section 2.5 is still very confusing and lacks structure. The reader does not know anything about this task at this point, so it is essential to first introduce the vigilance task per se in terms of the number slides in total and which slides are targets and which slides are not targets. In other words, what is the main thing that participants have to do in this task? Also, I could not understand what ISI referred to. For how long was each slide presented and what was the interval (if any) between the two consecutive slides? Once this is all explained, then the authors could add information about the cue words and explain also why these words are added (i.e., to induce task unrelated thoughts in participants). After this, the authors could explain everything in relation to thought probes. In fact, information about thought probes could precede the information about the cue words, if the authors think that this is a better way of presenting this information. But in any case, the task needs to be described step by step, in a logically structured way, so that the reader can immediately grasp the key aspects of the study instead of getting more and more confused and stopping at the end of each sentence.

The second relatively minor point concerns the paragraph on correlational analysis on sTUTs frequencies at Time 1 and Time 2. This paragraph I think should be presented as the last paragraph in this section starting with “Finally, the within-participants design allowed us…”. It was encouraging to see that there was such a strong correlation between the frequencies. However, can the authors please present the correlations separately for young and older adults to ensure that this is not a spurious correlation?

Some other minor points:

Line 304 – Reference number is 364. This can not be correct?

Line 341 – Please delete “given”

Line 471 – Please ensure that title of Table 2 indicates that means refer to spontaneous thoughts

Line 768 – Should be “and result from the activation of representations of previously constructed events”

7. PLOS authors have the option to publish the peer review history of their article (what does this mean?). If published, this will include your full peer review and any attached files.

Reviewer #1: No

Reviewer #2: No

---

## [Author Response · Author response to Decision Letter 1]

21 Jul 2020

Dear Dr. Faber,

We are submitting the revised manuscript PONE-D-20-07424R1, “The effects of aging and an episodic specificity induction on spontaneous task-unrelated thought”, for your consideration. We appreciate the positive evaluation of our review and the additional comments. 

In the present manuscript, we clarify the methods, in particular the vigilance task description, and the analyses, in particular the correlation between sessions 1 and 2, following the comments provided by Reviewer 2. 

Additionally, we made a slight correction to our method in the vigilance task, lines 303-304 in the manuscript with track changes, in which we deleted the word “motivated” as the scales here assessed concentration and difficulty only.

Please see the “Response to Reviewers” below for the full, point-by-point, description of our changes and clarifications. We hope you will find our modifications appropriate. 

Sincerely,

Magda Jordão

*

Response to Reviewers

Reviewer #2

Please note that the indications for lines in our replies refer to the manuscript with track changes

Overall, the revision reads very well and the authors have tried to address most of my comments and suggestions, except the one on the vigilance task. Despite the additions made, the description of the vigilance task in Section 2.5 is still very confusing and lacks structure. The reader does not know anything about this task at this point, so it is essential to first introduce the vigilance task per se in terms of the number slides in total and which slides are targets and which slides are not targets. In other words, what is the main thing that participants have to do in this task? Also, I could not understand what ISI referred to. For how long was each slide presented and what was the interval (if any) between the two consecutive slides? Once this is all explained, then the authors could add information about the cue words and explain also why these words are added (i.e., to induce task unrelated thoughts in participants). After this, the authors could explain everything in relation to thought probes. In fact, information about thought probes could precede the information about the cue words, if the authors think that this is a better way of presenting this information. But in any case, the task needs to be described step by step, in a logically structured way, so that the reader can immediately grasp the key aspects of the study instead of getting more and more confused and stopping at the end of each sentence.

R: We’ve now restructured the presentation of the task according to the suggestion (section 2.5, lines 267-349). After a brief mention of the motivation rating, that precedes the vigilance task, we describe what type and how many stimuli were presented, how long each were presented for, and what participants were asked to do. Then, we present the information related to the thought probes. After this, we add information about the cue words, including a mention to its role in inducing spontaneously retrieved thoughts. In the final paragraph of this section, we complement this information by providing additional detail about the reasoning behind the described features of the task.

The second relatively minor point concerns the paragraph on correlational analysis on sTUTs frequencies at Time 1 and Time 2. This paragraph I think should be presented as the last paragraph in this section starting with “Finally, the within-participants design allowed us…”. It was encouraging to see that there was such a strong correlation between the frequencies. However, can the authors please present the correlations separately for young and older adults to ensure that this is not a spurious correlation?

R: As suggested, we now presented this information as the last paragraph of the section and added the correlation information separately for younger and older adults, confirming that the correlation remains positive and significant for both groups (lines 528-533).

Some other minor points:

Line 304 – Reference number is 364. This can not be correct?

R: The reference number is 36, but in the track changes document this is not visible because the erased sign goes over the 4. This is visible in the previous manuscript without track changes (line 293) and in the present manuscript with track changes (line 281).

Line 341 – Please delete “given

R: Now deleted.

Line 471 – Please ensure that title of Table 2 indicates that means refer to spontaneous thoughts

R: Now added (line 489).

Line 768 – Should be “and result from the activation of representations of previously constructed events”

R: Yes, now corrected accordingly (line 785).

---

## [Editor Report · Decision Letter 2]

24 Jul 2020

The effects of aging and an episodic specificity induction on spontaneous task-unrelated thought

PONE-D-20-07424R2

Dear Dr. Jordao,

We’re pleased to inform you that your manuscript has been judged scientifically suitable for publication and will be formally accepted for publication once it meets all outstanding technical requirements.

Kind regards,

Myrthe Faber

Academic Editor

PLOS ONE
---

## [Editor Report · Acceptance letter]

30 Jul 2020

PONE-D-20-07424R2 

The effects of aging and an episodic specificity induction on spontaneous task-unrelated thought 

Dear Dr. Jordão:

I'm pleased to inform you that your manuscript has been deemed suitable for publication in PLOS ONE. Congratulations! Your manuscript is now with our production department. 

Kind regards, 

on behalf of

Dr. Myrthe Faber 

Academic Editor

PLOS ONE